# Contextually appropriate communication strategies for COVID-19 prevention in Kenya border regions: evidence from a mixed methods observational study in Busia and Mandera counties

Lydia Kaduka ![ORCID] ,[1,2] Joanna Olale,[3] Alexis Karamanos,[2] Joseph Mutai,[1] Clare Coultas,[4] Ismail Ahmed,[1] Veline L'Esperance,[5] Ursula Read,[6] Paola Dazzan,[7] John Kennedy Cruickshank,[2] Erastus Muniu,[1] Seeromanie Harding[2]

For numbered affiliations see end of article.

**Correspondence to**
Dr Lydia Kaduka;
lkaduka@kemri.go.ke

## ABSTRACT

**Objectives** Kenya has long and porous borders with its neighbouring countries. These regions, predominantly inhabited by highly mobile rural communities with strong cross-border cultural ties, present major challenges in managing movement of people and COVID-19 preventive measures. Our study sought to assess knowledge of COVID-19 prevention behaviours, how these varied by socioeconomic (SEC) factors and the challenges of engagement and implementation, in two border counties of Kenya.

**Methods** We conducted a mixed methods study using a household e-survey (Busia, N=294; Mandera, N=288; 57% females, 43% males), and qualitative telephone interviews (N=73: Busia 55; Mandera 18) with policy actors, healthcare workers, truckers and traders, and community members. Interviews were transcribed, English translated and analysed using the framework method. Associations between SEC (wealth quintiles, educational level) and knowledge of COVID-19 preventive behaviours were explored using Poisson regression.

**Results** Participants were mostly educated to primary school level (54.4% Busia, 61.6% Mandera). Knowledge of COVID-19 prevention varied by behaviour: hand washing-86.5%, use of hand sanitiser-74.8%, wearing a face mask-63.1%, covering the mouth when sneezing or coughing-56.3% and social distancing-40.1%. Differences in knowledge by area, educational level and the wealth index were marked, greatest for Mandera, the less educated and the poor. Interviews with stakeholders revealed challenges in health messaging, psychosocial and socioeconomic factors, lack of preparedness for truck border crossings, language barrier, denial and livelihood insecurity as key challenges to engagement with and implementation of COVID-19 prevention behaviours in the border regions.

**Conclusion** The influence of SEC disparities and border dynamics on knowledge and engagement with COVID-19 prevention behaviours calls for contextually appropriate risk communication strategies that are cognisant of community needs and local patterns of information flow. Coordinating response measures across border points

## STRENGTHS AND LIMITATIONS OF THIS STUDY

⇒ The study sampling strategy and high e-survey response rate (97%) were key strengths, which point to the versatility required in the conduct of research in emergency response with community engagement being central to the success of similar research projects.
⇒ The study did not objectively assess the knowledge of preventive behaviours. To overcome the potential bias of self-reported information, qualitative interviews were used to explore factors influencing the ability of the community to adhere to preventive measures.
⇒ Whereas Kenya's median age is 21, majority of the e-survey participants were>55 years. This group tends to be less tech-savvy and may explain the trust hierarchy observed in this study.
⇒ The study recorded a lower number of Mandera interviews. However, the context-specific challenges presented what may be considered socioculturally appropriate recommendations to inform future local efforts.

is crucial in winning communities' trust and maintaining essential economic and social activities.

## INTRODUCTION

Evidence continues to accrue on the dramatic global direct and indirect impacts of the COVID-19 pandemic. The pandemic has put a toll on both the social and economic structures of societies, contributing to global social, economic and health disparities.[1–3] Kenya has a population of ≈48 million, median age 21 years and 66.2% living on <US$3.20/day.[4] It has long and porous borders with Tanzania, Uganda, Ethiopia, South Sudan and Somalia, predominantly inhabited by highly mobile rural communities with strong cross-border

cultural ties. Eight weeks after the first reported COVID-19 case in Kenya on 13 March 2020, a quarter of all 781 reported cases came from these border counties.[1] A single-wave epidemic peaked within 100–200 days after the first reported case.[3] Recognising its vulnerability to COVID-19 due to its position as a major travel and transport hub and relatively weak health system, Kenya implemented several mitigation measures including: a dusk to dawn curfew; mandatory mask wearing; prohibition of large public gatherings; and closure of borders.[5 6] Indirect impacts of the measures resulted in increased poverty levels, disrupted livelihoods, deepened inequalities and decelerated progress towards the Sustainable Development Goals.[7–9]

The needs of communities at border regions are an important consideration for Kenya and its neighbours due to their shared vision of regional health security. The porous borders in the Mandera triangle (Kenya, Ethiopia and Somalia borders) and Busia (Kenya–Uganda border) limit the ability of partner states to monitor and identify active cases during pandemics.[10–15] With only 18% of the Kenyan population fully vaccinated by 19 January 2022,[16] the mainstay of the response remained adoption of preventive behaviours to reduce transmission.

In this study, we aimed to assess the levels of knowledge of key COVID-19 prevention behaviours, and whether these varied by socioeconomic factors, and obtained qualitative narratives to illuminate contextual factors impacting on engagement with prevention measures in Busia and Mandera border counties of Kenya. These two border counties continue to record poorer health outcomes due to long-standing issues in accessing reasonable quality healthcare because of poor health infrastructure, remoteness from the capital city and high poverty rates. They are also among the eight lowest ranked counties in Kenya with the least healthcare workers per capita.[17]

## METHODS

### Study design

A convergent parallel mixed methods study design was used to collect both quantitative and qualitative data in the two counties between September and October 2020. The design allows concurrent collection and analysis of both data sets, and convergence at interpretation stage.[18]

### Study settings

Mandera county located in Northern Kenya shares a border of 861 km with Ethiopia to the North and 682 km with Somalia to the East (figure 1). It is located 1052 km from Kenya's capital city Nairobi, covers an area of 25 798 km², has a population of 867 457, population density of 33 persons/km², high poverty level (77.6%) and is inhabited predominantly by Muslim Somalis. Mandera's main economic activities are pastoralism and cross-border trade. Of Mandera's 775 085 population aged>3 years, 72% have never been to a learning institution. Busia county located 475 km from Nairobi, borders Lake Victoria to the North and Uganda to the West (total border distance, 772 km). With a 1700 km² area and population of 893 000, population density is 527 persons/km² and poverty level is 69%. Its main economic activities are cross-border trade, fishing and agriculture and is inhabited predominantly by Christians. Overall, 12.5% of Busia's 820 788 population aged>3 years have never been to school.[19]

### Quantitative survey

#### Sample size and sampling procedure

A cross-sectional household e-survey targeting adults aged>18 years was undertaken in September 2020. Sample size calculation was based on a prevalence of 50% for the indicator of interest (knowledge) and a maximum relative SE for the estimated proportion of 7% to give a sample size of 205. Assuming 20 participants per

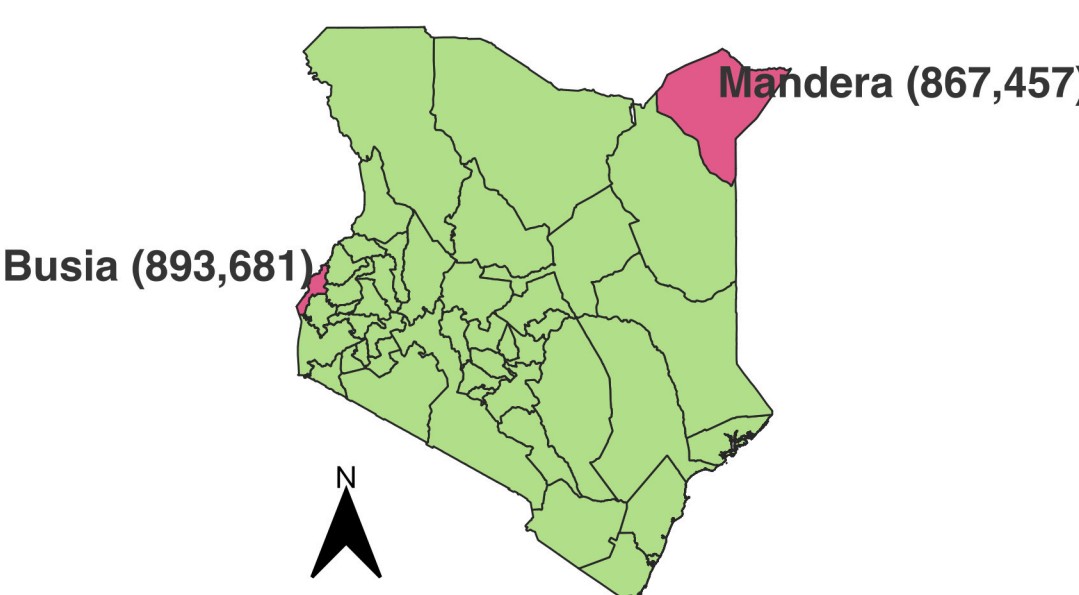

**Figure 1** Map showing the location of Busia and Mandera counties in Kenya.

cluster (communities), an intraclass correlation coefficient between clusters of 0.01, and a non-response rate of 10%, the resulting sample size was 269 rounded off to 300 participants, equivalent to 15 clusters per county. A three-stage sampling process was used. The first stage involved listing of all public health facilities in the study areas (Busia 66; Mandera 79) and the square root allocation method used to determine the health facilities sampled per area. Fifteen facilities per area were selected using systematic sampling with a random start. For each selected health facility, one community health unit (CHU) served by the health facility was selected using simple random sampling. Community health workers (CHWs) in selected CHUs assisted in developing the household sampling frame. The second stage involved sampling 20 households within CHUs using systematic sampling with a random start. The third stage involved respondent selection in sampled households using the Kish Grid method[20] as a way of balancing representation.

### Data collection and analysis

The e-survey data was collected using the WHO COVID-19 Rapid Quantitative Assessment Tool (online supplemental file 1) adapted to the local context.[21] The study used the REDCap software[22] to automate and share the survey tool with study participants. The e-questionnaire was administered either electronically on a smartphone or via telephone interview for participants with literacy or technology challenges. The study worked with trained CHWs in verifying respondent's eligibility and access to a smartphone with internet connection. A REDCap link was shared that allowed the respondent to consent, fill and submit the e-survey. Due to erratic internet in the study sites, the participants were allowed to submit their survey forms any time within the day using a unique return code generated specific to each participant to safeguard completion. Technical support was offered via WhatsApp.

### Outcome measures

Knowledge about COVID-19-related preventive health behaviours was assessed by asking survey participants the following question 'Do you know how to prevent COVID-19?', to which they responded with a yes/no on the following individual statements; prevent COVID-19 by (1) wearing a face mask, (2) keeping a distance of 2 m from others, (3) covering the mouth and nose when sneezing or coughing, (4) using a hand sanitiser and (5) washing hands with soap and water regularly. Affirmative responses were categorised as the reference category. Exposure status was assessed by asking participants if they had experienced any of the following symptoms—fever, cough, shortness of breath, sore throat, fatigue, loss of taste or smell, or eye infection, 2 weeks prior to the survey.

### Correlates and confounders

Socioeconomic circumstances (SEC) were measured using education and the wealth index, two commonly used SEC indices in Kenya.[19] The wealth index was calculated using data on household's ownership of 14 selected assets (online supplemental table 1). The weights (factor scores) for each of the assets were generated through principal components analysis. Years of education and adult education were categorised as following: (1) secondary education and above (more than 9 years of compulsory years of education), (2) upper primary education (5–8 years of compulsory education), (3) lower primary education (1–4 years of compulsory education) or/and Islamic education (Duksi or Madrasa) and (4) no education.

Participant's sex, age (categorised in 4 age bands 18–24, 25–54, 55–65 and 65+) and survey location (Mandera/Busia) were considered as potential confounders. Trusted sources of information, desire for additional information on COVID-19 and perception of risks stemming from SARS-CoV-2 were considered as potential mediators and were assessed by asking participants to respond with a yes/no to each category for the following questions: 'Which source/who do you trust the most to receive information related to the new Coronavirus?' with response categories (1) the radio, (2) the TV, (3) the WhatsApp messenger, (4) CHWs and (5) religious leaders. Desire for additional information on COVID-19 was measured using the question: 'What more would you like to know about COVID-19?' with response categories (1) symptoms of the new Coronavirus, (2) how it is transmitted, (3) what to do if you have symptoms; Perceived risks from COVID-19 (1) 'How dangerous do you find the new Coronavirus?', (2) 'Do you think you are likely to become sick with the new Coronavirus?' and (3) 'Do you think any of your family members are at risk of becoming sick from the new Coronavirus?'. Responses to the first question were categorised as very dangerous, dangerous or 'not at all'. The latter two were combined due to small numbers reporting 'not at all'. Response to the other two questions consisted of yes, no and do not know. Information on gender, survey location and age group were also collected.

### Statistical analyses

All statistical analyses used the Stata V.16 software[23] and the statistical significance level set at 5%. The association between the exposures (wealth quintiles and educational level) and the outcomes of interest (knowing that COVID-19 can be prevented by specific behaviours) was explored using Poisson regression.[24] Poisson regression was preferred over logistic regression to allow relative risks to be presented, rather than ORs, which are often misrepresented. All regression analyses applied survey weights to account for the higher proportion of women included in the survey (57% vs 43% for males) and the higher proportion of survey participants with secondary education and above (28.5% vs 23.7% for those with no education). The multivariable regression analysis adjusted for wealth quintiles, educational level, gender, survey location, age group, trusted sources of information about COVID-19, desired additional information on COVID-19 and perception of risks.

## Qualitative interviews

### Sampling procedure

Purposive sampling was used to identify and select respondents with professional or personal experience of COVID-19. Semistructured, in-depth telephone interviews were done with policy-makers, healthcare workers, truckers, traders and COVID-19 survivors and carers to assess their experiences and needs in relation to the pandemic and mitigation measures specific to border counties' populations. Interviewees were mapped using an initial stakeholders list, which included the counties' leadership, health management teams and the COVID-19 response teams, to which further additions were made through snowballing. Interviews were completed between September and October 2020.

### Data collection and analysis

In-depth interviews were carried out to explore understanding and experiences of the pandemic (online supplemental file 2). The interviews were conducted in either Kiswahili or English based on preference of the interviewees, time range of 30–60 min per interview. Questions related to the following topics were explored: community response to and experience of the pandemic; health systems' preparedness and response to the pandemic; and border-specific issues related to the pandemic (only data pertaining to prevention behaviours are presented in this paper). The recorded interviews were transcribed, translated and checked for quality by the Kenyan research team and a UK-based coauthor (CC) who is fluent in Swahili. The transcripts were then analysed using the framework method by Gale *et al*:[25] open-coding of the data was undertaken in pairs or small teams (with at least one Kenyan and one UK-based individual) so that sociocultural differences in interpretations of the data could be identified and engaged with. The team then met altogether to develop an analytical framework with key categories (ie, groupings of codes that are clearly defined). The analytical framework was then applied to all of the transcripts. Illustrative quotes were charted into a framework matrix and the data was interpreted and organised into themes.

### Patient and public involvement

The research question was arrived at on interrogating the existing COVID-19 response measures and realising a potential gap in addressing the needs of border communities. The county administration assessed the relevance of the proposed research and gave permission for community participation in the study. CHWs and village elders assisted in sensitising and mobilising community participation, participated in mapping and selection of households, and sampling of study participants. The study findings were discussed in stakeholder forums with representation from the counties, community-based organisations and members of health committees of the local legislature. Recommendation items actionable at county level were taken up by respective county teams for implementation.

## RESULTS

### Knowledge of key COVID-19 prevention behaviours in Busia and Mandera

We assessed levels of knowledge of key COVID-19 prevention behaviours, and whether these varied by socioeconomic factors. Table 1 shows the sample profiles for the 582 adults (Busia: 50.5%; Mandera 49.5%) who participated in the e-survey.

Majority (95%) of the participants were 25 years and above, 14% lacked formal education. Socioeconomic disadvantage was high in both locations, with just over a quarter in Busia and over half of those in Mandera reporting having received only primary 1–4 years, Duksi or Madrasa education. Mandera was consistently more disadvantaged than Busia across all measures. Overall, 17% reported a fever and cough, while 4% reported loss of smell/taste 2 weeks before the survey.

In both areas, the most known common prevention behaviours identified were washing hands regularly 86.5% (Busia 95.9%, 95% CI 92.3 to 97.9; Mandera 77%, 95% CI 70.4% to 82.5%) and use of a hand sanitiser 74.8% (Busia 84.4%, 95% CI 78.7% to 88.9%; Mandera 65.2%, 95% CI 58.3% to 71.6%) and the least commonly known was standing 2 m away from others 40.1% (Busia 52%, 95% CI 45.3% to 59.1%; Mandera 28%, 95% CI 22.2% to 34.6%). Significantly lower proportions in Mandera than in Busia reported knowledge of prevention behaviours, having had secondary education, being in the most disadvantaged fourth and fifth wealth quintiles, having trust in COVID-19-related information from various sources including the radio and TV and having received COVID-19-related knowledge on symptoms, transmission and what to do if they have symptoms. There were some notable differences for trusted sources of COVID-19 information. In Busia, 93% cited the radio as the most trusted source, whereas in Mandera the radio (55%), CHWs (44%) and religious leaders (46%) shared the top ranking for trusted source. Table 2 shows the influence of adjusting for SEC, trusted sources of information, additional information and perception of risk on the Mandera–Busia differences in knowledge of prevention behaviours.

The lower knowledge of washing hands in Mandera compared with Busia remained after fully adjusting, with hardly any shift in the risk ratios. The differences between Mandera and Busia in knowledge about covering the mouth when coughing/sneezing and wearing a mask were no longer significant on adjustment for SEC, for using a sanitiser on adjustment for trusted sources of COVID-19 information, and for keeping a 2 m distance from others on adjustment for all confounders.

Based on the pooled Busia and Mandera sample and after adjusting for confounders including location, the knowledge of prevention behaviours did not vary by education (online supplemental figure 1). In contrast,

**Table 1** Sample profiles in Busia and Mandera counties

| | Busia (N=294) | Mandera (N=288) | All (N=582) |
|---|---|---|---|
| | Proportion (95% CI) | Proportion (95% CI) | Proportion (95% CI) |
| **Age group** | | | |
| 18–24 (early working age) | 8.6 (5.6 to 12.9)* | 1.4 (0.6 to 3.5) | 5.0 (3.4 to 7.3) |
| 25–54 (prime working age) | 6.8 (4.3 to 10.6) | 7.2 (4.7 to 10.8) | 7.0 (5.1 to 9.5) |
| 55–64 (mature working age) | 59.9 (52.8 to 66.7) | 52.1 (45 to 59.1) | 56.0 (51 to 60.9) |
| 65+ (elderly) | 24.7 (18.5 to 32)* | 39.3 (32.3 to 46.8) | 32 (27.2 to 37.3) |
| **Education level** | | | |
| Secondary and above | 40.9 (34.4 to 47.6)* | 15.1 (11.4 to 19.9) | 28.0 (24.0 to 32.3) |
| Primary 5–8 years | 28.6 (23.6 to 34.1)* | 3.4 (2 to 5.8) | 16.0 (13.4 to 19.0) |
| Primary 1–4 years/Duksi/Madrasa | 25.8 (19.1 to 34)* | 58.2 (51.6 to 64.5) | 42.0 (36.9 to 47.3) |
| None | 4.7 (3.0 to 7.4)* | 23.3 (18.9 to 28.3) | 14.0 (11.6 to 16.8) |
| **Wealth quintiles** | | | |
| Highest quintile | 16.9 (2.5 to 12.6)* | 23.2 (17.8 to 29.7) | 20.1 (16.5 to 24.2) |
| Second | 21.5 (2.7 to 16.7)* | 12.7 (9.1 to 17.4) | 17.1 (14.0 to 20.7) |
| Third | 29.7 (3.4 to 23.4)* | 16.3 (11.6 to 22.4) | 23.0 (18.9 to 27.7) |
| Fourth | 17.8 (2.7 to 13.1)* | 18.4 (14 to 23.9) | 18.1 (14.8 to 22.0) |
| Lowest | 14 (2.4 to 10)* | 29.4 (23.1 to 36.6) | 21.7 (17.8 to 26.2) |
| **Preventative behaviours** | | | |
| Wash hands regularly with soap and water | 95.9 (92.3 to 97.9)* | 77 (70.4 to 82.5) | 86.5 (82.6 to 89.6) |
| Cover mouth and nose when coughing or sneezing | 64.3 (57.3 to 70.7)* | 48.3 (41.3 to 55.3) | 56.3 (51.4 to 61.1) |
| Hand sanitiser use | 84.4 (78.7 to 88.9)* | 65.2 (58.3 to 71.6) | 74.8 (70.3 to 78.9) |
| Standing 2 m away from other people | 52.2 (45.3 to 59.1)* | 27.9 (22.2 to 34.6) | 40.1 (35.4 to 45.0) |
| Wear a face mask | 70.9 (64.2 to 76.9)* | 55.3 (48.2 to 62.3) | 63.1 (58.2 to 67.9) |
| **Trusted source of information on COVID-19** | | | |
| Radio | 92.8 (88.6 to 95.5)* | 54.8 (47.8 to 61.7) | 73.8 (69.4 to 77.8) |
| TV | 51.6 (44.7 to 58.4)* | 11.8 (7.9 to 17.4) | 31.7 (27.3 to 36.4) |
| WhatsApp | 25.2 (19.8 to 31.5) | 19.1 (14.2 to 25.3) | 22.2 (18.4 to 26.5) |
| Community health workers | 56.4 (49.4 to 63.1)* | 43.7 (37.0 to 50.7) | 50 (45.1 to 55) |
| Religious leaders | 33.3 (27.3 to 39.9)* | 46.2 (39.3 to 53.3) | 39.7 (35 to 44.6) |
| **Kind of information received** | | | |
| Symptoms of COVID-19 | 82.3 (76.5 to 86.9)* | 60.3 (53.2 to 66.9) | 71.3 (66.6 to 75.5) |
| How SARS-CoV-2 is transmitted | 74 (67.5 to 79.5)* | 40.9 (34.2 to 47.9) | 57.4 (52.5 to 62.2) |
| What to do if you have symptoms | 65.7 (58.9 to 72)* | 28.7 (22.7 to 35.6) | 47.2 (42.3 to 52.2) |
| **Perception of risk** | | | |
| **How dangerous COVID-19 is?** | | | |
| Very dangerous | 89.3 (84.4 to 92.8)* | 73.2 (66.7 to 78.9) | 81.3 (77.2 to 84.8) |
| Less dangerous/not dangerous | 10.7 (7.2 to 15.6)* | 26.8 (21.1 to 33.3) | 18.7 (15.2 to 22.8) |
| **Do you think any of your family member are at higher risk of becoming sick?** | | | |
| Yes | 53.3 (46.4 to 60)* | 20.4 (15.6 to 26.2) | 36.8 (32.3 to 41.7) |
| No | 13.8 (9.8 to 19.2) | 13.8 (9.9 to 18.9) | 13.8 (10.9 to 17.4) |
| Don't know | 32.9 (26.8 to 39.6)* | 65.8 (59.1 to 71.9) | 49.4 (44.4 to 54.3) |
| **Do you think you are likely to become sick with COVID-19?** | | | |
| Yes | 56.1 (49.2 to 62.8)* | 16.3 (12.2 to 21.4) | 36.2 (31.6 to 41) |

**Table 1** Continued

| | Busia (N=294) | Mandera (N=288) | All (N=582) |
|---|---|---|---|
| | Proportion (95% CI) | Proportion (95% CI) | Proportion (95% CI) |
| No | 14 (9.9 to 19.3) | 16.8 (12.3 to 22.6) | 15.4 (12.2 to 19.2) |
| Don't know | 29.9 (24.1 to 36.6)* | 66.9 (60.3 to 72.9) | 48.4 (43.5 to 53.4) |
| COVID-19 symptoms | | | |
| Fever | 12.9 (9 to 18.1) | 20.2 (15.6 to 25.8) | 16.5 (13.4 to 20.2) |
| Cough | 12.5 (8.6 to 17.7) | 21.1 (16.3 to 26.9) | 16.8 (13.6 to 20.6) |
| Shortness of breath | 7.7 (4.6 to 12.5) | 9.9 (6.6 to 14.6) | 8.8 (6.4 to 12) |
| Sore throat | 7.1 (4.3 to 11.7) | 6.4 (3.8 to 10.3) | 6.7 (4.7 to 9.6) |
| Fatigue | 5.2 (2.8 to 9.5) | 4.5 (2.5 to 9.1) | 4.9 (3.1 to 7.4) |
| Loss of taste or smell | 5.9 (3.2 to 10.7) | 2.1 (0.9 to 5) | 4 (2.4 to 6.6) |

*95% CIs do not overlap.

regardless of location, those in the most disadvantaged fifth wealth quintile remained less likely to report knowledge of 3 of the 5 prevention behaviours—washing hands regularly (−20%, 95% CI −7% to −31%), covering mouth and nose when sneezing or coughing (−34%, 95% CI −10% to −53%) and standing two metres away from others (−39%, 95% CI −4% to −61%).

Significant interactions were observed between quintiles of wealth and location for knowing that covering the mouth when coughing or sneezing (F=35.23, p<0.001) and keeping 2 m distance (F=13.26, p=0.009) could prevent COVID-19. The most socioeconomically disadvantaged participants in Mandera were 62% (95% CI 33% to 79%) less likely to know that COVID-19 could be prevented by covering their mouth while coughing/sneezing and 61% (95% CI26% to 89%) less likely to know that COVID-19 could be prevented by keeping 2 m distant (see online supplemental figure 2). No significant differences in the knowledge of these behaviours were observed between

the least and the most socioeconomically disadvantaged in Busia.

### Qualitative insights into factors impacting on engagement with prevention and control measures

A total of 73 qualitative interviews (Busia: n=55; Mandera: n=18) were carried out with policy actors (n=8), healthcare workers (n=34), truckers and traders (n=7), COVID-19 survivors and carers (n=8) and other community members including religious leaders, village elders, persons living with disability and commercial sex workers (n=16) in the two counties. Age of interviewees ranged between 23 and 74 years.

The qualitative interviews provided insights into contextual factors which impacted on engagement with prevention and control measures. Table 3 provides a summary, including quotes from the qualitative interviews, to illustrate factors impacting on engagement with prevention measures in the two border regions. Lack of trust in the

**Table 2** The influence of adjusting for socioeconomic circumstances, trusted sources of COVID-19 information and additional information and perception of risk on the Mandera–Busia differences in knowledge of prevention behaviours

| | Mandera–Busia difference in prevention behaviour, adjusted for gender and age | Education level and wealth quintiles† | Trusted sources of information about COVID-19 knowledge† | Desired additional information on COVID-19 and perception of risk† |
|---|---|---|---|---|
| Wash hands regularly with soap and water | 0.80 (0.74 to 0.87)*** | 0.81 (0.74 to 0.89)*** | 0.81 (0.73 to 0.90)*** | 0.86 (0.78 to 0.96)** |
| Cover mouth and nose when coughing or sneezing | 0.72 (0.61 to 0.87)*** | 0.81 (0.64 to 1.03) | 0.87 (0.67 to 1.14) | 0.99 (0.76 to 1.28) |
| Hand sanitiser use | 0.77 (0.68 to 0.86)*** | 0.80 (0.69 to 0.92)** | 0.88 (0.75 to 1.02) | 0.99 (0.85 to 1.16) |
| Standing 2 m away from other people | 0.52 (0.40 to 0.68)*** | 0.61 (0.44 to 0.85)** | 0.58 (0.40 to 0.84)** | 0.71 (0.50 to 1.01)+ |
| Wear a face mask | 0.77 (0.66 to 0.90)** | 0.97 (0.79 to 1.20) | 0.93 (0.73 to 1.17) | 1.06 (0.83 to 1.36) |

Risk ratio (95% CI), Busia is the reference category.
*p<0.05, **p<0.01, ***p<0.001.
†p<0.1and >0.05

**Table 3** Qualitative Insights on impacts on adherence to prevention measures

| Themes | Categories | Illustrative quotes |
|---|---|---|
| Distrust of national government and politicians | Politicians not adhering to COVID-19 restrictions | BM-Imam-118: "They look at the politicians and how they are conducting themselves. Those breaking the rules do so because those that are supposed to encourage them are not adhering to the rules"<br>BM-Comm.HealthVolunteer-018: "people see the leaders walking without wearing masks [… so] they believe there is no corona"<br>MM-Trader-074 "we see politicians doing things without social distance… it only means there is no COVID" |
| | Media coverage of misappropriation of COVID-19 funds | BM-Comm.HealthVolunteer-003: "having heard that there is a lot of money that was squandered through Kenya Medical Supplies Authority… majority of them are not doing the right thing [i.e. taking preventative measures]"<br>BM-PublicHealthOfficer-052: "people started relaxing in adhering after this scandal was unearthed whereby, they say that billions of money was stolen" |
| Health messaging not adequately contextualised to local communities | Insufficient attention to impact of isolation on livelihoods | BM-COVIDSurvivor-111: "people fear to go and test, because they know you will just dump somebody there [in an isolation centre] for ten days […] you don't know the person, the way they eat, survive, he is just there like that in the name of quarantine, isolation, all those things" |
| | Inadequate community engagement and tailored public health messaging | BM-Policyactor-087: "we need to have more community engagement, with interpersonal communication kind of approach, so that you do more than just general messages, you are doing specific messages to specific groups of people, truck drivers, sex workers, maybe clearing agents and so on"<br>BM-Policyactor-113: "Communication on protection was not well done. We needed more of community engagement and good communication to ensure that all issues are clarified through question and answers"<br>BM-Policyactor-096: "remember the kind of information which was being given out initially [… led] by the Ministry of Health was more of. well it was a caution but it really sounded to the public like this thing is going to wipe away so many people […] the words were really confusing, quarantine, isolation and so on" |
| | Circulation of misinformation in the community | —'COVID is just like malaria, it should not be feared' (BM-Farmer-085; BF-Comm. HealthVolunteer-004).<br>—'COVID only affects people in big cities and truckdrivers'/'people do not know people who have had COVID - seeing is believing' (BF-Policyactor-025; BM-CommercialSexWorker-068; MM-Policyactor-007). |
| | Difficulties engaging with pastoralist communities (in Mandera) | MM-Policyactor-112: "the pastoralist nature of our society, you know people moving from here to the other side [of the border], uncontrollable has also been a serious issue [… they get health information from listening to] vernacular radio stations […] [but] compliance is another thing"<br>MM-Policyactor-007: "we had cases that were imported from Somalia by people who went to sell their livestock [and crossed the border unchecked]" |
| Psychosocial and socioeconomic impacts on adherence to preventive measures | Desire for spiritual and social connection | BM-Policyactor-087: "people are social they want to be together but now they have been told to keep a distance… And when churches were closed, this was very strange… you will find it is affecting people spiritually"<br>BM-Policyactor-087: "where people used to greet you by shaking hands, and now people are supposed to learn a new mode of greeting which is not our culture and to kick Kenyan culture is not so easy"<br>MF-COVIDSurvivor-098: "To protect yourself from corona, you are supposed to wear masks, don't greet people and also restrain from hugging people but sometimes you forget and greet people using your hands, you hug people because you are used to this" |
| | Socioeconomic barriers | BM-PLWDisability-117: "where will we disabled people get masks from? […] We have to work hard and buy for ourselves from our own pockets, buy for the children"<br>MM-Healthworker-037: "there's nobody who can provide them with masks, they were buying at […] twenty shillings or fifty shillings, and so it was very expensive?'"<br>MM-Policyactor-112: "There are these small [informal] businesses [e.g. tea/coffee stands], and… when we want to enforce those [social distancing] regulations it becomes very difficult because this is somebody's daily bread". |
| | Religious and social spaces supporting adherence | MM-Trader-074: "Even at the shop I have sanitiser, running water with soap, so people when they come to the shop they sanitize".<br>MF-COVIDSurvivor-098: "We have [hand] washing stations in hospitals, madrasa, at the mosque"<br>MM-PublicHealthOfficer-039 "we had a meeting with all the religious leaders… [to discuss how some preventive] measures… So after sensitizing, at least people have started accepting [preventive measures], because it was a big challenge for people to accept social distance in the place of worship" |

Continued

**Table 3** Continued

| Themes | Categories | Illustrative quotes |
|---|---|---|
| Issues specific to border areas | Lack of preparedness for truck border crossings | BM-Policyactor-096: "initially we had lots of challenges… the trucks piled up… [resulting] in drivers not being able to cross over to Uganda on time, they ended up mingling with the rest of the population and so we knew it was just a matter of time before we got the first community cases"<br>BM-PublicHealthOfficer-052: ""you find this community transmission tends to be growing [because truck drivers must wait for 2–3 days for the COVID test results]… we feel that there is a need for testing to be done across the entire [border] township […] but we lack the resources for doing this" |
| | Health prevention information not translated into all needed languages | BM-Policyactor-087: "people from Congo, from Rwanda, they speak French and […] most of our communications are in Kiswahili and English and therefore—like now we don't have any material for French here at the border and that is a huge challenge" |
| | Lack of consistency in cross-border communication | MM-Policyactor-007: "previous cooperation about polio issues […] can also help in responding to this pandemic"<br>BM-PublicHealthOfficer-052: "It is compulsory in Kenya to put on masks…these drivers when in Uganda they are not allowed to mingle with people, they think that anybody coming from Kenya is having Corona [because of the mask wearing]"<br>BM-Policyactor-113: "Kenya and Uganda might not have been reading from the same script. The countries should have the same lab where they collect the samples and the same approach to the communities. I believe they have community health issues at the border points [like us]. In times of a pandemic, no one is to blame. We have our own challenges but those that spread across East Africa would be dealt better with collaboration" |
| | Disruptions to cross-border trade and food security | BM-TraditionalHealer-072: "you can't cross the border to go to the Ugandan side [of the forest to collect herbs], if you are found, maybe you passed sideways [informal crossing], you get arrested. We had to stop working"<br>BM-Comm.HealthVolunteer-003: "there is food insecurity, people used to do farming freely, do business, they are unable to cross the border" |

(B/M) indicates location (Busia/Mandera), (M/F) indicates gender (male/female).

national government and political leaders was noted as a key barrier to engagement with prevention measures in the two border regions. Participants reported that this lack of trust mainly stemmed from alleged misappropriations of funds and politicians not adhering to COVID-19 restrictions.

Other reported impacts included socioeconomic factors such as the loss of livelihoods (especially in the informal sector), and increased cost of living due to rising food prices occasioned by disruptions in local trade and food imports, which impacted COVID-19 prevention measures. Psychosocial factors such as the centrality of spiritual support and family social connections in communities' cultures and way of life conflicted with preventive measures.

Additionally, there was reported to be inadequate attention to providing targeted, context specific, accessible risk communication and public health advice which impacted on equitable understanding and adherence. Challenges included use of technical global pandemic language and procedures that were not translatable to the local setting; lack of community engagement in tailoring communication messages; lack of targeted messaging for the pastoralist community in Mandera; language barriers with French-speaking truckers from Congo and Rwanda; and misinformation within the community regarding COVID-19 symptoms and transmission. Some initiatives by the two county governments were reported to facilitate engagement with prevention, including informal

provisions of hand washing stations at the border posts, markets, shops and religious institutions.

Challenges specific to border areas included, lack of preparedness for truck border crossings which led to pooling of truckers at the border especially in Busia, thus risking community transmission. Cross-border collaboration in enforcing the preventive measures was enhanced through leveraging an already existing infectious disease surveillance network in Mandera but challenged by stigmatisation of truckers in Busia.

## DISCUSSION

We assessed the levels of knowledge of key COVID-19 prevention behaviours in the two border counties. The best-known preventive behaviour was washing of hands regularly and the least known was standing two metres apart, with Mandera significantly disadvantaged. We observed disparate knowledge of COVID-19 prevention measures based on socioeconomic status. Those with less education and those in lower socioeconomic groups were less likely to adhere to COVID-19 prevention measures. Key factors impacting on engagement with COVID-19 prevention measures included lack of trust in the leadership, deficits in public health communication, livelihood insecurity, prioritisation of social support over quarantine/social distancing and border-specific complexities such as language barriers and congregating of truckers due to lack of preparedness for truck border crossings.

## Influence of socioeconomic inequalities on knowledge of COVID-19 prevention behaviours

Our study findings are consistent with previous studies in Kenya and other parts of Africa that observed variations in knowledge of COVID-19 prevention measures based on employment status, access to information, education level, geographical location and gender.[26–28] A study in coastal Kenya[29] reported 60% knowledge of COVID-19 prevention measures, citing inequalities in knowledge, awareness and reach of health information.

The high awareness of COVID-19 reported in this study can largely be attributed to adoption of a whole of government and multi-agency approach (policy and technical) in response to the COVID-19 threat, accompanied by widespread media campaigns and news reports. A long history and experience with infectious disease response may have made messaging on hand hygiene and cough/sneeze etiquette more acceptable and readily implementable in Kenya. This, however, was not achieved with wearing masks and maintaining social distance as they were considered novel interventions. Lags in supply and affordability of masks and sanitising agents, and difficulties in sustaining lockdowns and movement restriction as reported in this study and elsewhere,[14] contributed to failure in observing regular use of masks and social distancing. Low adherence to mask wearing and social distancing at a time of widespread consensus on COVID-19 being an airborne disease coupled with the low vaccination rate,[16] call for innovative ways to engage communities in developing contextually relevant public health measures, particularly when introducing novel interventions, cognisant of the influence of local culture and the impact of the socioeconomic environment on the capacity for adherence. Likewise, emphasis on building community knowledge and not just providing top–down instructions, engaging communities in the adaptation processes as previously shown in Bangladesh,[30] has the potential to improve adherence and ownership through better understanding. Further, the reported need for additional information on the disease indicates a lag in adjusting the communication messaging to the evolution of the pandemic, a gap that requires addressing.

## Public health information

Our study found that community engagement with preventive measures improved with access to appropriate messages from preferred channels of communication. Socioeconomic inequalities observed were slightly mitigated by trust placed in the source of information (religious leaders, WhatsApp, radio and CHWs), signalling the opportunity to capitalise on the trust hierarchy to facilitate adherence to preventive behaviours. The pandemic communication strategy needs to be tailored to the local context,[31] and the dynamics of information flow to address gaps in information penetration and engagement. Key in this process is the translation of the standardised pandemic language and procedures to local contexts through participatory processes. In addition, the reported community's loss of trust in the genuineness of the government efforts warrants a concerted effort to win back the public to comply with COVID-19 prevention, possibly through dialogue. The pastoralist communities require special attention as a key population due to their nomadic way of life that can marginalise them in respect of public health awareness and prevention measures and could increase the risk of cross-border transmission and potential recurrent outbreaks.[15] The risk is exacerbated by limited access to water and sanitation. There is need therefore for concerted effort in improving health literacy through tailored education campaigns and messaging in the border regions, and especially among the pastoralists, supported by structural interventions to enable disadvantaged and marginalised communities adhere to prevention measures.

An added dimension to risk communication is the stability and strength of mobile network coverage, which tends to be poorer in rural areas. Representativeness in sampling of participants in both counties could reflect the relative ease of access to local radio when compared with television and internet-based social media, in contrast to other work done in urban settings in Kenya.[26 27]

## Border dynamics

Border health security is key to safeguarding the East African Community's economic and social security. Studies show that implementation of COVID-19 interventions is likely problematic for populations with weak health systems, high density housing, porous borders and insecurity.[32] The disruptions to trade and livelihoods have had profound effects on border communities.[9] Concerns over financial stability (loss of income and food security) have persisted over time. There is a need to economically empower and build resilience among border communities to withstand the current and similar shocks in future. The border counties had to cater for a bigger population than their official figures, and as such, budgetary allocation to these counties should recognise this reality. The mandatory testing of truck drivers caused a major backlog in Busia which lies on the Northern Transport Corridor. As gathering points for large numbers of people, border crossings are inherently a risk point for transmission of SARS-CoV-2. This calls for better organisation in designating and preparing holding areas in border regions. This could have potentially protected the local communities in the initial phase of COVID-19 and later with established community spread, the uninfected drivers.

In addition to the harmonised testing protocol of truckers by Kenya and her neighbours, there is also a need to strengthen cross-border surveillance networks to support contact tracing to increase the likelihood of containment of outbreaks and reduce the risk posed by cross-border health threats.[33] Adherence to prevention measures and movement restrictions at the borders was reportedly undermined by porous borders and strong social ties between the cross-border communities. The COVID-19 preventive measures and mitigation measures

should thus be accompanied by intensive community engagement to minimise a sense of alienation among these communities. The use of informal crossings should be mitigated by developing criteria accessible to all that recognises the inherent economic and social challenges in the region.

## Study limitations

The study did not objectively assess the knowledge of preventive behaviours, nor sought participants' self-reported levels of adherence. To overcome the potential bias of self-reported information, qualitative interviews were used to explore factors influencing the ability of the community to adhere to preventive measures, which would be useful going forward than adherence reported at a single time point. Adopting standardised international research tools enables comparison of findings even though the tools do not comprehensively capture all possible sources of information, reflecting an issue of adapting Western-designed surveys. Whereas Kenya's median age is 21, majority of the e-survey participants were >55 years. This group tends to be less tech-savvy. To minimise the digital divide and barriers to participant participation, the study trained CHWs, deployed a 'light' data collection tool and adaptive workflows and provided robust technical support. The study recorded a lower number of Mandera interviews due to the geographical expansiveness of the area. However, the context-specific challenges presented what may be considered sociocul-turally appropriate recommendations to inform future local efforts.

## CONCLUSION

This research highlights the importance of pandemic response and communication in engagement with COVID-19 preventive measures in border settings. As in other contexts, socioeconomic inequalities were reflected in knowledge of and adherence to COVID-19 prevention measures, with the less educated and those in lower socio-economic groups at greatest risk. The influence of trusted sources of information in mitigating such inequalities demonstrate the need for tailored public health communication strategies that are cognisant and responsive to local context and patterns of information flow. Further effort needs to be made in risk communication by modifying health messaging in tandem with evolution of the pandemic and incorporating community communication needs. Improving health literacy may support better health outcomes for the population.

Livelihood insecurity directly prevents the most disadvantaged from adhering to prevention measures. Economic empowerment and building resilience against future shocks is thus an important investment the country needs going forward. Coordinating response measures across border points is crucial in winning the community's trust and maintaining essential economic and social activities.

**Author affiliations**
¹Centre for Public Health Research, Kenya Medical Research Institute, Nairobi, Kenya
²Faculty of Life Sciences & Medicine, School of Life Course Science, King's College London, London, UK
³Centre for Clinical Research, Kenya Medical Research Institute, Nairobi, Kenya
⁴School of Medical Education, King's College London, London, UK
⁵School of Population Health and Environmental Sciences, King's College London, London, UK
⁶Warwick Medical School, University of Warwick, Coventry, UK
⁷Institute of Psychiatry Psychology and Neuroscience, King's College London, London, UK

**Acknowledgements** We are grateful to all the study participants from Busia and Mandera counties who took time to participate in this study. We thank the Director KEMRI, the National Commission for Science, Technology and Innovation, and the County Governments of Busia and Mandera for granting permission and providing an enabling environment to undertake this study. Many thanks to our research collaborators at Kings College London for their technical guidance and support, and the KEMRI County Cluster Coordinators Mr. Surrow Adow and Ms. Olipher Makwaga for their support. We thank our research assistants Schiller Mbuka, Rodgers Ochieng, Melvine Obuya, Esther Andia, Miriam Bosire, Doreen Mitaru and Priscilla Maiga for their tireless efforts in bringing the study to fruition.

**Contributors** LK, JO, AK, JM, CC, IA, VL, UR, PD, JKC, EM and SH participated in conceiving the study, implementation and drafting of the manuscript. All authors contributed to the interpretation and subsequent revision of the paper. All authors approved the final version of the manuscript. LK is responsible for the overall content as guarantor.

**Funding** This work was supported by the Kenya Medical Research Institute (KEMRI) Internal Research Grant number KEMRI/COV/SPEC/002. SH, JKC, UR funded by MR/N015959/1. SH, AK, LK funded by MR/S009035/1. SH funded by NIHR7\1004. VL was funded by DRF 2017-10-132. SH and PD were also funded by MR/R022739/1. PD funded by NIHR203318.

**Map disclaimer** The inclusion of any map (including the depiction of any boundaries therein), or of any geographic or locational reference, does not imply the expression of any opinion whatsoever on the part of *BMJ* concerning the legal status of any country, territory, jurisdiction or area or of its authorities. Any such expression remains solely that of the relevant source and is not endorsed by *BMJ*. Maps are provided without any warranty of any kind, either express or implied.

**Competing interests** None declared.

**Patient and public involvement** Patients and/or the public were involved in the design, or conduct, or reporting, or dissemination plans of this research. Refer to the Methods section for further details.

**Patient consent for publication** Not applicable.

**Ethics approval** This study involves human participants. The study protocol was approved by the KEMRI Scientific and Ethics Review Committee (Ref. No. KEMRI/SERU/CPHR/4021) and the National Commission for Science, Technology and Innovation (Ref. No. 864492). Participants gave informed consent to participate in the study before taking part.

**Provenance and peer review** Not commissioned; externally peer reviewed.

**Data availability statement** All data relevant to the study are included in the article or uploaded as online supplemental information.

**ORCID iD**
Lydia Kaduka http://orcid.org/0000-0001-8746-0533

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
