## [Reviewer comments · BMJ Open]

ARTICLE DETAILS

TITLE (PROVISIONAL)	Contextually appropriate communication strategies for COVID-19 prevention in Kenya border regions: Evidence from a mixed methods observational study in Busia and Mandera Counties.
AUTHORS	Kaduka, Lydia; Olale, Joanna; Karamanos, Alexis; Mutai, Joseph; Coultas, Clare; Ahmed, Ismail; L'Esperance, Veline; Read, Ursula; Dazzan, Paola; Cruickshank, John; Muniu, Erastus; Harding, Seeromanie

VERSION 1 – REVIEW

REVIEWER	Odhiambo, J University of Nairobi, Mathematics
REVIEW RETURNED	05-May-2022

GENERAL COMMENTS	Add these articles to the literature review:  1. Odhiambo, J., Weke, P., & Ngare, P. (2020). Modeling Kenyan economic impact of corona virus in Kenya using discrete-time Markov chains. Journal of Finance and Economics, 8(2), 80-85. 2. Odhiambo, J. O., Ngare, P., Weke, P., & Otieno, R. O. (2020). Modelling of covid-19 transmission in kenya using compound poisson regression model. Journal of Advances in Mathematics and Computer Science, 101-111. 3. Odhiambo, J., Okungu, J., & Mutuura, C. (2020). Stochastic modeling and prediction of the COVID-19 spread in Kenya. Eng. Math, 4, 31-35. 4. Naryongo, R., Onyango, J., Njagi, L., & Nakirya, M. Modeling of COVID-19 Transmission under Markov Chains in Uganda. 5. Odhiambo, J., Ngare, P., & Weke, P. (2022). Bühlmann credibility approach to systematic mortality risk modeling for sub-Saharan Africa populations (Kenya). Research in Mathematics, 9(1), 2023979. These will make your literature review much stronger especially from the already done researches in the area.
---

REVIEWER	Harada, Kouji Kyoto University, Graduate School of Medicine
REVIEW RETURNED	21-May-2022

GENERAL COMMENTS	This study was conducted in September 2020. The information is outdated compared with current situation in COVID pandemic. Authors should state the importance of the study objective and results. Authors conducted this study neat border areas in Kenya. But the questionnaire did not include specific questions related in this issue. In addition, there was no comparison with other areas in non-border
---

	areas. Hence, the research questions in this study cannot be answered. This is an e-survey but the method was not described. E-survey cannot be delivered to those who have no e-devices. The population should be biased especially in economically poor population. It does not meet with the study objective. COVID-19 infection is usually mediated by droplets and aerosols. Hand hygiene is not important. How do authors use current results for future measurement of COVID prevention? In quantitative survey, the item in questionnaire is not clear. Authors asked participants on behaviors and information source. But availability of them was not confirmed. If the participants had the device, the answer could be biased. Similarly, desired information does not mean the insufficiency of them. If the information was enough, the answer would be 'no'. Authors did not provide the process to make the questionnaire. Also, for qualitative survey, the semi-structured questions should be related with the quantitative survey, but the no rationale was not provided. From the results of interviews, there was little relationship with individual prevention behaviors. There was no advantage in a mixed method. Less educated and low SEC population had less knowledge. This was not surprising. What was the new and specific findings in this area? Authors did not provide any specific answer to improve the knowledge in this area. Poverty is difficult to be solved more than COVID-19.
--	--

REVIEWER	Kawale, Paul African Institute for Development Policy Malawi Office
REVIEW RETURNED	22-Dec-2022

GENERAL COMMENTS	Thank you for the opportunity to review your manuscript. This is an important topic, and well done for the good science. I would like to make a few suggestions for your consideration as follows: 1. Introduction  - The choice of the two boarder regions is well-justified. For context, it may be enlightening to also know the relative Covid-19 infection rates, either at the study sites or other appropriate cluster (eg county). - It would also be good to know the relative rates of adherence to health prevention measures, either for the study counties or Kenya, compared to other counties or countries. - I appreciate your citation of Pinchoff et al. to mention socio-economic interactions (effects) of Covid-19 in Kenya. However, some literature on socio-economic CAUSES of Covid-19 infection prevention knowledge (or indeed, Kenyans' general disease prevention behaviors) would do well to justify your choice of socio-economic variables. 2. Methods  - Mixed-methods was an appropriate design, though it is unclear how the quantitative and qualitative approaches "converge" to inform
---

each other across the study's research questions.

- Thank you for citing the National Census to justify the age brackets and choice of variables. However, it is not cited how the age bracket 25-54 was determined. Is it some standard or law? Other studies?

- It is not indicated in the ethics section whether informed consent was oral or written.

- It is interesting that you selected two counties that are mutually unique in terms of their populations' religion and education levels. Was religion one of the factors considered or adjusted for during analysis, assuming Christians and Moslems may have different hygiene routines?

- Being an e-survey, the sampling procedure is appropriate, and the community leaders' assistance to identify the n'th respondent is impressive. However, it is not indicated whether the tool was self-administered or not, and whether access to the e-survey skewed your respondent representation since they would need access to smartphone.

- There is need for evidence to justify the choice of the five Covid-19 information source categories. Was this derived from the qualitative interviews, for example? Literature?

- It appears this is an observational study. However, I was wondering whether there was a theoretical framework or model that informed particularly the qualitative analysis.

3. Results

- Seeing that respondents' ages ranged from 14, it is not indicated in the ethics section whether assent was sought for the minors.

- A large proportion of study respondents are in the upper and second wealth quantiles, while methods section presents the two border counties as very poor. Further, your analytical methods indicate that you had to adjust for the higher proportion of respondents with secondary education and above. At the same time, you indicate in your results that the study participants have low socioeconomic status. What was the cut-off quantile? Did this bias or skew your sample's representatives of the border counties?

- You have a very elaborate quantitative analysis of the differences between the two counties. It would be interesting to also present these differences in the qualitative analysis, either to support or contrast the quantitative findings.

4. Discussion

- It seems knowledge and practice ("adherence") are being used interchangeably. Did the study measure knowledge, practice or both? If both, then the two must be presented and discussed separately.

- You cite other studies identifying gender as one of the key socioeconomic variables to Covid-19 knowledge. Would it be possible to identify gender differences in your study? I noticed that gender and age were clumped together during analysis.

- Some of the issues discussed are not supported by the data. For instance, there is mention of pastoralists, mobile network coverage, budgetary allocation, access to water/sanitation or internet-based social media, yet there is no mention of them in the results.

- The discussion is clear and mentions many things that are rather "obvious", and misses the opportunity to discuss factors from the study that are unique to border regions. For instance, the qualitative findings identify spiritual, religious and social spaces as socioeconomic factors, while quantitative findings identify religious leaders as one of the trusted information sources, and the methods describe the different dominant religious communities (Moslem vs

	Christian) between the two counties. There is no discussion of faith or religion as one of the socioeconomic variable determining Covid-19 prevention knowledge and/or practice. 5. Conclusion - Again the conclusion is well-written, but generally repeats “general” knowledge that reads more like a social media post than key conclusions from the data, focusing on the unique situation at borders that are different from other settings. Thank you once again for the opportunity to review your manuscript. It is well-written, with only minor spelling errors (eg you wrote “Whats Up” instead of “WhatsApp”). Once the suggested revisions described above have been clarified, this is a worthy paper to be published.
--	---

VERSION 1 – AUTHOR RESPONSE

Reviewer: 1

Dr. J Odhiambo, University of Nairobi

Comments to the Author:

Add these articles to the literature review:

1. Odhiambo, J., Weke, P., & Ngare, P. (2020). Modeling Kenyan economic impact of corona virus in Kenya using discrete-time Markov chains. *Journal of Finance and Economics*, 8(2), 80-85.
 2. Odhiambo, J. O., Ngare, P., Weke, P., & Otieno, R. O. (2020). Modelling of covid-19 transmission in kenya using compound poisson regression model. *Journal of Advances in Mathematics and Computer Science*, 101-111.
 3. Odhiambo, J., Okungu, J., & Mutuura, C. (2020). Stochastic modeling and prediction of the COVID-19 spread in Kenya. *Eng. Math*, 4, 31-35.
 4. Naryongo, R., Onyango, J., Njagi, L., & Nakirya, M. Modeling of COVID-19 Transmission under Markov Chains in Uganda.
 5. Odhiambo, J., Ngare, P., & Weke, P. (2022). Bühlmann credibility approach to systematic mortality risk modeling for sub-Saharan Africa populations (Kenya). *Research in Mathematics*, 9(1), 2023979.
- These will make your literature review much stronger especially from the already done researches in the area.

***Comment from the Editor: You do not have to use citations suggested in the review, but please try to answer the concern.

Please also see attached comments

Response: We take note of the modelling approaches to COVID-19 transmission and recommendations covered in the five articles, key among them being the need for Kenya counties to tailor COVID-19 response measures to their contexts. These sentiments have been captured in the discussion section (Page 17-18).

Reviewer: 2

This study was conducted in September 2020. The information is outdated compared with current situation in COVID pandemic. Authors should state the importance of the study objective and results.

Response: We appreciate the concern raised regarding relevance of the study findings. It is worth noting that apart from COVID-19, the East African region continuous to experience outbreaks of other infectious disease such as Ebola and Chikungunya. Emerging evidence show socioeconomic factors, population structures, misinformation, disregard of basic public health precautions, failure to draw upon behavioural and social sciences, and failure to collaborate (Ref

31) as some of the key drivers of disease transmission and poor epidemic control globally. Our study lends weight by providing context/border specific insights to these issues to inform future tailoring of strategies for effective response.

Authors conducted this study near border areas in Kenya. But the questionnaire did not include specific questions related in this issue. In addition, there was no comparison with other areas in non-border areas. Hence, the research questions in this study cannot be answered.

Response: The study was designed to assess and inform response measures in the highly porous border regions of Kenya given the significance of border control in surveillance, prevention and control of infectious disease transmission. Border specific barriers and facilitators to engagement with the preventive measures are provided in Table 3. The study questionnaire is also provided in Supplementary File 1.

This is an e-survey but the method was not described. E-survey cannot be delivered to those who have no e-devices. The population should be biased especially in economically poor population. It does not meet with the study objective.

Response: The e-survey details have been provided on page 6 as follows “The e-questionnaire was administered either electronically on a smartphone or via telephone interview for participants with literacy or technology challenges. The study worked with trained community health workers (CHWs) in verifying respondent’s eligibility and access to a smart phone with internet connection. A REDCap link was shared which allowed the respondent to consent, fill and submit the e-survey. Due to erratic internet in the study sites, the participants were allowed to submit their survey forms any time within the day using a unique return code generated specific to each participant to safeguard completion. Technical support was offered via WhatsApp.”

COVID-19 infection is usually mediated by droplets and aerosols. Hand hygiene is not important. How do authors use current results for future measurement of COVID prevention?

Response: The utility of the current findings serves and goes beyond COVID-19 prevention. As stated earlier, the findings will help inform tailored response to infectious diseases. Some of the recommendations proposed may help in building solidarity in community response to COVID-19 and other equally infectious diseases, which requires understanding the role of individual behaviour on transmission, and is secured by consistent community engagement, risk communication and appropriate messaging.

In quantitative survey, the item in questionnaire is not clear. Authors asked participants on behaviors and information source. But availability of them was not confirmed. If the participants had the device, the answer could be biased.

Response: Details on administration of the questionnaire are provided in the response above and in the main text (Page 6). One of the stated study limitations (Page 19) is the potential bias of self-reported data. To overcome this limitation, the mixed methods research approach was used to explore factors that influence adherence to prevention measures.

Similarly, desired information does not mean the insufficiency of them. If the information was enough, the answer would be ‘no’.

Response: The question on desire for additional information “What more would you like to know about the disease?” as shown in Supplementary File 1, part 3.17 provided the option of an alternative response, which in this case could have been reported as none.

Authors did not provide the process to make the questionnaire. Also, for qualitative survey, the semi-structured questions should be related with the quantitative survey, but the no rationale was not provided.

Response: Details of quantitative data collection process using the WHO tool (Supplementary File 1), automation and how it was administered have been included in the main text (Page 6). The qualitative tool (Supplementary File 2) also provides the prompts used in exploring the contextual factors, supporting the quantitative arm. Similarly the rationale for the qualitative work is included in the main text (Page 8).

From the results of interviews, there was little relationship with individual prevention behaviors. There was no advantage in a mixed method.

Response: The quantitative and qualitative findings converge in the discussion section where

qualitative findings shed light on potential drivers of the socioeconomic disparities impacting adherence to preventive measures. The first paragraph of the discussion section (Page 17) highlights key factors impacting engagement.

Less educated and low SEC population had less knowledge. This was not surprising. What was the new and specific findings in this area? Authors did not provide any specific answer to improve the knowledge in this area. Poverty is difficult to be solved more than COVID-19.

Response: Factors such as disruptions to trade, livelihoods and socio-cultural ties, communication barriers, and pooling of truckers at border points were some of the border specific factors (Page 17) that impacted adherence to COVID-19 preventive measures, and possibly exacerbated the inequalities observed.

Reviewer: 3

Thank you for the opportunity to review your manuscript. This is an important topic, and well done for the good science. I would like to make a few suggestions for your consideration as follows:

1. Introduction

- The choice of the two border regions is well-justified. For context, it may be enlightening to also know the relative Covid-19 infection rates, either at the study sites or other appropriate cluster (eg county).

Response: Information on the case burden and patterns has been included in the introduction section (Page 4).

- It would also be good to know the relative rates of adherence to health prevention measures, either for the study counties or Kenya, compared to other counties or countries.

Response: Emerging information on adherence and practice of the preventive measures in other counties has been included in the discussion section (Page 17).

- I appreciate your citation of Pinchoff et al. to mention socio-economic interactions (effects) of Covid-19 in Kenya. However, some literature on socio-economic CAUSES of Covid-19 infection prevention knowledge (or indeed, Kenyans' general disease prevention behaviors) would do well to justify your choice of socio-economic variables.

Response: Also included in the introduction is a citation by Brand et al., *Science* 374, 989–994 (2021) (Ref 3) which lends weight to the influence of socioeconomic factors in viral transmission.

2. Methods

- Mixed-methods was an appropriate design, though it is unclear how the quantitative and qualitative approaches “converge” to inform each other across the study’s research questions.

Response: Details of the study design by Creswell & Pable-Cark, 2011 have been included in the text (Page 5).

- Thank you for citing the National Census to justify the age brackets and choice of variables. However, it is not cited how the age bracket 25-54 was determined. Is it some standard or law? Other studies?

Response: Definition of prime working age is provided for by the International Labour Organization, and recognized in Kenya (<https://ilostat ilo.org/resources/concepts-and-definitions/description-labour-force-statistics/>)

- It is not indicated in the ethics section whether informed consent was oral or written.

Response: E-consenting was done. Details have been included on page 9. E-consenting was done on REDCap, which involved mandatorily displaying the informed consent document. At the end of the consent document the participant would click a mandatory check box in which they either provided or withdrew consent. The survey was only accessible upon consent. The digital record that the box was checked was archived as evidence that consent was obtained.

- It is interesting that you selected two counties that are mutually unique in terms of their populations' religion and education levels. Was religion one of the factors considered or adjusted for during analysis, assuming Christians and Moslems may have different hygiene routines?

Response: Adjustments were done for wealth quintiles, educational level, gender, survey location, age group, trusted sources of information which includes religious leaders, desired additional information on COVID-19 and perception of risks (Page 8).

- Being an e-survey, the sampling procedure is appropriate, and the community leaders' assistance to identify the nth respondent is impressive. However, it is not indicated whether the tool was self-administered or not, and whether access to the e-survey skewed your respondent representation since they would need access to smartphone.

Response: Details on administration of the e-questionnaire and how we mitigated challenges in access to smart phones and/or internet connectivity have been included on Page 6.

The questionnaire was administered either electronically on a smartphone or via telephone interview for participants with literacy or technology challenges. The study worked with trained community health workers (CHWs) in verifying respondent's eligibility and access to a smart phone with internet connection. A REDCap link was shared which allowed the respondent to consent, fill and submit the e-survey. Due to erratic internet in the study sites, the participants were allowed to submit their survey forms any time within the day using a unique return code generated specific to each participant to safeguard completion. Technical support was offered via WhatsApp.

- There is need for evidence to justify the choice of the five Covid-19 information source categories. Was this derived from the qualitative interviews, for example? Literature?

Response: The study used the WHO COVID-19 Rapid Quantitative Assessment Tool adapted to the local context (Supplementary File 1).

- It appears this is an observational study. However, I was wondering whether there was a theoretical framework or model that informed particularly the qualitative analysis.

Response: Analysis of the qualitative work was guided by the framework method (Page 8) (Reference 25: Gale NK, Heath G, Cameron E, et al. Using the framework method for the analysis of qualitative data in multi-disciplinary health research. BMC Medical Research Methodology 2013; 13:117)

3. Results

- Seeing that respondents' ages ranged from 14, it is not indicated in the ethics section whether assent was sought for the minors.

Response: Apologies for the typo error in Table 1. The study involved adult participants aged 18 years and above as described on Page 7.

- A large proportion of study respondents are in the upper and second wealth quintiles, while methods section presents the two border counties as very poor. Further, your analytical methods indicate that you had to adjust for the higher proportion of respondents with secondary education and above. At the same time, you indicate in your results that the study participants have low socioeconomic status. What was the cut-off quantile? Did this bias or skew your sample's representatives of the border counties?

Response: This is attributed to weighting of the dataset, which among others, helps address biases that may occur when undertaking such population studies.

- You have a very elaborate quantitative analysis of the differences between the two counties. It would be interesting to also present these differences in the qualitative analysis, either to support or contrast the quantitative findings.

Response: The unique responses to each of the border regions are highlighted in Table 3 and results section (Page 14)

4. Discussion

- It seems knowledge and practice ("adherence") are being used interchangeably. Did the study measure knowledge, practice or both? If both, then the two must be presented and discussed separately.

Response: This is well noted and corrected. The study assessed knowledge of COVID-19 prevention behaviours.

- You cite other studies identifying gender as one of the key socioeconomic variables to Covid-19

knowledge. Would it be possible to identify gender differences in your study? I noticed that gender and age were clumped together during analysis.

Response: The outcomes of the adjustments for SEC are provided in Table 2 in the text.

- Some of the issues discussed are not supported by the data. For instance, there is mention of pastoralists, mobile network coverage, budgetary allocation, access to water/sanitation or internet-based social media, yet there is no mention of them in the results.

Response: Mandera's main economic activity is pastoralism and cross-border trade (Page 5), availability of handwashing stations (Table 3) alludes to sanitation issues. The statement on budgetary allocation relates to the resource constraints reported in Table 3. Issues to do with network coverage and social media in the discussion section are referenced and in support of issues pertaining to risk communication in the results section.

- The discussion is clear and mentions many things that are rather "obvious", and misses the opportunity to discuss factors from the study that are unique to border regions. For instance, the qualitative findings identify spiritual, religious and social spaces as socioeconomic factors, while quantitative findings identify religious leaders as one of the trusted information sources, and the methods describe the different dominant religious communities (Moslem vs Christian) between the two counties. There is no discussion of faith or religion as one of the socioeconomic variable determining Covid-19 prevention knowledge and/or practice.

Response: Under the public health information sub-topic in the discussion section, the potential of trusted sources of information in mitigating the inequalities and promoting adherence has been captured (Page 18). Socio-economic inequalities observed were slightly mitigated by trust placed in the source of information (religious leaders, WhatsApp, radio, and community health workers), signaling the opportunity to capitalize on the trust hierarchy to facilitate adherence to preventive behaviours.

5. Conclusion

- Again the conclusion is well-written, but generally repeats "general" knowledge that reads more like a social media post than key conclusions from the data, focusing on the unique situation at borders that are different from other settings.

Response: The conclusion highlights the major challenges reported from the two border regions and potential mitigation measures to inform planning and guard against future shocks.

Thank you once again for the opportunity to review your manuscript. It is well-written, with only minor spelling errors (eg you wrote "Whats Up" instead of "WhatsApp"). Once the suggested revisions described above have been clarified, this is a worthy paper to be published.

Response: Well noted. This has been corrected in Table 1.

VERSION 2 – REVIEW

REVIEWER	Harada, Kouji Kyoto University, Graduate School of Medicine
REVIEW RETURNED	10-Mar-2023
GENERAL COMMENTS	The authors did not provide any information on representativeness of the samples even in revised manuscript. If this is a random sample in this area and little effect from digital divide, basic demographics should be compared with statistics in Table 1. Response rate was 97%. This is unusually high. If the authors provide incentives to the participants, it should be noted.

VERSION 2 – AUTHOR RESPONSE

Response to reviewer's comments

Thank you for the review comments. Below is the response to the two comments raised.

1. The authors did not provide any information on representativeness of the samples even in revised manuscript. If this is a random sample in this area and little effect from digital divide, basic demographics should be compared with statistics in Table 1.

Response: Additional information on the three-stage sampling process has been added on page 6 as follows "The first stage involved listing of all public health facilities in the study areas (Busia 66; Mandera 79) and the square root allocation method used to determine the health facilities sampled per area. Fifteen facilities per area were selected using systematic sampling with a random start. For each selected health facility, one community health unit (CHU) served by the health facility was selected using simple random sampling. Community health workers (CHWs) in selected CHUs assisted in developing a household sampling frame. The second stage involved sampling 20 households within CHUs using systematic sampling with a random start. The third stage involved respondent selection in sampled households using the Kish Grid method [20] as a way of balancing representation".

2. Response rate was 97%. This is unusually high. If the authors provide incentives to the participants, it should be noted.

Response: The positive response is attributed to the partnership between the research team and the community. The study leveraged on the vibrant community strategy in Kenya and worked closely with the community health workers in the selected CHUs in developing the sampling frame, identifying sampled households and eligible participants. Secondly, the research subject was on a pertinent issue that everyone was concerned and enthusiastic about.